# Circulating Tumor DNA as a Preoperative Marker of Recurrence in Patients with Peritoneal Metastases of Colorectal Cancer: A Clinical Feasibility Study

**DOI:** 10.3390/jcm9061738

**Published:** 2020-06-04

**Authors:** Jamie J. Beagan, Nina R. Sluiter, Sander Bach, Paul P. Eijk, Stijn L. Vlek, Daniëlle A. M. Heideman, Miranda Kusters, D. Michiel Pegtel, Geert Kazemier, Nicole C. T. van Grieken, Bauke Ylstra, Jurriaan B. Tuynman

**Affiliations:** 1Department of Pathology, Amsterdam UMC, Vrije Universiteit Amsterdam, Cancer Center Amsterdam, De Boelelaan 1117, 1081 HV Amsterdam, The Netherlands; j.beagan@amsterdamumc.nl (J.J.B.); pp.eijk@amsterdamumc.nl (P.P.E.); dam.heideman@amsterdamumc.nl (D.A.M.H.); d.pegtel@amsterdamumc.nl (D.M.P.); nct.vangrieken@amsterdamumc.nl (N.C.T.v.G.); 2Department of Surgery, Amsterdam UMC, Vrije Universiteit Amsterdam, Cancer Center Amsterdam, De Boelelaan 1117, 1081 HV Amsterdam, The Netherlands; n.sluiter@amsterdamumc.nl (N.R.S.); s.bach@amsterdamumc.nl (S.B.); s.vlek@amsterdamumc.nl (S.L.V.); m.kusters@amsterdamumc.nl (M.K.); g.kazemier@amsterdamumc.nl (G.K.); j.tuynman@amsterdamumc.nl (J.B.T.)

**Keywords:** circulating tumor DNA, colorectal cancer, peritoneal metastases, CRS-HIPEC, droplet digital PCR, liquid biopsy

## Abstract

Cytoreductive Surgery and Hyperthermic Intraperitoneal Chemotherapy (CRS-HIPEC) may be curative for colorectal cancer patients with peritoneal metastases (PMs) but it has a high rate of morbidity. Accurate preoperative patient selection is therefore imperative, but is constrained by the limitations of current imaging techniques. In this pilot study, we explored the feasibility of circulating tumor (ct) DNA analysis to select patients for CRS-HIPEC. Thirty patients eligible for CRS-HIPEC provided blood samples preoperatively and during follow-up if the procedure was completed. Targeted Next-Generation Sequencing (NGS) of DNA from PMs was used to identify bespoke mutations that were subsequently tested in corresponding plasma cell-free (cf) DNA samples using droplet digital (dd) PCR. CtDNA was detected preoperatively in cfDNA samples from 33% of patients and was associated with a reduced disease-free survival (DFS) after CRS-HIPEC (median 6.0 months vs median not reached, *p* = 0.016). This association could indicate the presence of undiagnosed systemic metastases or an increased metastatic potential of the tumors. We demonstrate the feasibility of ctDNA to serve as a preoperative marker of recurrence in patients with PMs of colorectal cancer using a highly sensitive technique. A more appropriate treatment for patients with preoperative ctDNA detection may be systemic chemotherapy in addition to, or instead of, CRS-HIPEC.

## 1. Introduction

Metastatic colorectal cancer (CRC) remains the second most common cause of cancer-related death, despite improvements in treatment over recent decades [1]. Peritoneal metastases (PMs) are diagnosed in 10–25% of CRC patients [2,3,4], either at the time of primary tumor diagnosis or during subsequent investigations. If restricted to the peritoneum, referred to as isolated PMs, treatment with systemic chemotherapy confers a median overall survival (OS) of 12–18 months [5,6,7,8]. Patients with isolated PMs that are limited in their spread throughout the peritoneum may qualify for Cytoreductive Surgery and Hyperthermic Intraperitoneal Chemotherapy (CRS-HIPEC) treatment, which is associated with a disease-free survival (DFS) of 13.1 months and an increased median OS of 35–45 months [9,10,11,12]. Patients with the least extensive peritoneal spread have the potential to experience the greatest benefit from CRS-HIPEC, reflected in a median OS of 56 months [13]. The addition of HIPEC following CRS did not show significant survival benefit in the recent large randomized-controlled PRODIGE-7 trial [14]. However, unlike the patients included in our study, patients in the PRODIGE-7 trial had received neo-adjuvant systemic therapy prior to CRS-HIPEC and had received oxaliplatin rather than mitomycin C during HIPEC [14]. Nevertheless, CRS performed in high-volume expert centers resulted in an OS of 41 months [14]. This demonstrates that a dedicated centralized multimodality approach to treatment offers improved survival for patients with PMs.

Unfortunately, shortcomings in modalities of detection and quantification of PMs often result in diagnosis at an advanced stage, when the peritoneal spread is extensive. These patients typically see reduced benefit from CRS-HIPEC because of a higher incidence of recurrence after the procedure [15,16] and because they experience greater treatment-related morbidity from extensive CRS [17]. Timely preoperative detection of PMs could help decide if any potential survival benefit outweighs the treatment-related morbidity. Similarly, recurrence of PMs after CRS-HIPEC is a common phenomenon [13] that also tends to be diagnosed at an advanced stage. Improved detection in this situation could indicate an early start to chemotherapy to reduce disease symptoms, or a repeat CRS-HIPEC with potentially curative intent if the recurrence is isolated and limited [18].

If PMs are not detected during resection of the primary tumor, initial detection is by subsequent (PET)-CT imaging, which is limited to a sensitivity of 72% and further decreases to 11% for nodules smaller than 5 mm [19]. Carcinoembryonic antigen (CEA) levels can be elevated in the blood of CRC patients, particularly in those with metastases. However, the sensitivity of the CEA test is limited in patients with isolated PMs [20,21]. If PMs are suspected, diagnostic laparoscopy (DLS) is often performed preoperatively to determine the Peritoneal Cancer Index (PCI), a numeric score ranging from 1–39 that combines lesion size with the number of affected abdomino-pelvic regions [10,22]. Although DLS offers a higher sensitivity and specificity for detection, it is invasive and often still underestimates the extent of PMs [23]. The most reliable measure of PMs is an intraoperative assessment performed immediately before the intended CRS-HIPEC. However, up to 25% of patients are disqualified at this point due to irresectable PMs, reflected by a PCI higher than 20, so undergo an open–close procedure whereby the abdomen is closed without CRS-HIPEC [24]. There is an urgent clinical need for less invasive and more accurate tools to detect and quantify the extent of PMs. Novel approaches include improved imaging techniques such as diffusion-weighted (DW) MRI [25] and liquid biopsy analysis.

Recent technical advances have enabled circulating tumor (ct) DNA, the fraction of cell-free (cf) DNA that originates from tumor cells, to be detected in plasma with high sensitivity and specificity. Analysis can serve as a dynamic marker in CRC patients by the quantification of cfDNA levels and the identification of tumor-specific genetic and epigenetic markers including; mutations, structural variations and methylation [26,27,28]. Studies that used digital PCR methods have shown ctDNA to be detectable in up to 100% of CRC patients with systemic metastases and in 73% of those without evidence of systemic metastasis [29]. However, in a study that used a next-generation sequencing (NGS) panel to test patients with resectable PMs of diverse cancer types, ctDNA was only detectable preoperatively in 39% of cases [30]. The lower plasma ctDNA representation in CRC patients with isolated PMs compared to systemic metastases is probably due to inherent biological differences in ctDNA shedding.

The detection and quantification of relatively low quantities of ctDNA in plasma from CRC patients with PMs requires a highly sensitive technique. Droplet digital (dd) PCR has emerged as one of the most sensitive and specific methods of ctDNA analysis in the oncology setting [31]. By specifically targeting genomic loci in a tissue-guided manner, ddPCR can precisely quantify DNA fragments that contain either the mutant or wild-type nucleotide in a relatively simple, fast and cost-effective work-flow. Although this approach has previously been applied to colorectal malignancies [32,33], to our knowledge, it has not been used as a biomarker in a well-defined cohort of CRC patients who are candidates for CRS-HIPEC.

In this pilot study, we aimed to explore the feasibility of ctDNA analysis in a clinical situation to select patients for CRS-HIPEC. We assessed the capability of ctDNA to quantify the extent of PMs and its suitability as a preoperative prognostic marker of recurrence.

## 2. Experimental Section

### 2.1. Study Design and Patients

Patients diagnosed with synchronous or metachronous PMs of colorectal adenocarcinoma and considered eligible for CRS-HIPEC following standard work-up including imaging and DLS, were initially included in this study at the Cancer Center Amsterdam of Amsterdam University Medical Center (location VUmc) between August 2016 and March 2018 [11,34]. Patients were preoperatively excluded from the study if: the estimated extent of PMs was deemed to be irresectable by subsequent (PET-) CT or DLS; systemic metastases were detected (excluding resectable liver metastases with minimal tumor burden) [35]; or PMs removed during a previous procedure were found by histological assessment to be non-colorectal in origin or were not adenocarcinoma. Patients were excluded from the study on a technical basis if no mutations were identified in PMs. Clinical and pathological data were retrospectively obtained from patient records (Appendix A) [10,22]. The mismatch repair status of PMs was not tested in any of the patients included in the study. The outcome of the CRS was determined according to the maximal size of residual tumor tissue and was classified as: R1) when no macroscopically visible tumor remained in situ (complete resection), R2a) when the residual tumor was smaller than 2.5 mm or R2b) when it was larger than 2.5 mm [36]. CRS-HIPEC was performed according to a standard protocol [11,37]. If a complete CRS was achieved, HIPEC was performed using the open coliseum technique with mitomycin C. If residual tumor tissue remained after CRS, HIPEC was not performed [6,34].

### 2.2. Ethics Approval and Consent to Participate

This study was registered with the Dutch Trial Registry [38] and was conducted in accordance with the Declaration of Helsinki with the approval of the Amsterdam UMC, VU University Medical Ethical Testing Committee (2016.254-NL57226.029.16 and 2017-302(A2018). All patients provided written informed consent to participate in the study.

### 2.3. Blood and Tumor Tissue Collection

PMs previously removed alongside primary tumor resection were retrieved from the Biobank at Amsterdam University Medical Centers (UMC) -location VUmc, and retrospectively analyzed to identify mutations to test in cfDNA. Preoperative blood samples were collected after patients were placed under general anesthesia but immediately before surgical incision for the intended CRS-HIPEC procedure. Blood was kept at room temperature until plasma separation within 8 h of collection (Appendix A). At least one postoperative blood sample was taken by venipuncture from all CRS-HIPEC patients, typically within 2–4 weeks, but no later than 3 months after the procedure. Further samples were taken during routine follow-up every 3 months, up to 25 months after CRS-HIPEC. If a recurrence was diagnosed during follow-up by physical assessments and (PET-) CT imaging, an additional blood sample was taken at diagnosis or within 1 month. No postoperative blood samples were obtained from patients who did not undergo the complete CRS-HIPEC procedure because the presence of residual tumor excluded them from postoperative ctDNA analysis.

### 2.4. DNA Isolation and Mutation Analysis

Formalin-fixed paraffin-embedded (FFPE) tumor tissue of PMs was processed as previously described [39,40]. Genomic DNA was subsequently isolated using a Qiagen QIAamp DNA FFPE Tissue kit according to the instructions of the manufacturer (Qiagen, Venlo, the Netherlands). Cell-free DNA was isolated from up to 3 mL aliquots of plasma using the Qiagen QIAsymphony Circulating DNA Kit. A plasma sample of known cfDNA concentration was included in each isolation run to ensure consistent performance of the isolation kit. Genomic DNA from PMs underwent NGS-based mutation analysis using the TruSeq Amplicon Cancer Panel (TSACP; Illumina Inc., San Diego, CA, USA) or a High-Resolution Melting assay followed by Sanger sequencing (HRM-sequencing) (Appendix A) [40,41]. Only genomic variants of known oncogenic significance identified by HRM-sequencing, or by TSACP sequencing with a Variant Allele Frequency (VAF) of ≥3% in the PMs were included (Appendix A). These mutations were targeted in cfDNA samples using specific mutant and wild-type ddPCR primer and probe combination kits (BioRad, California, USA) (Appendix A). Results for each assay were used to calculate the VAF and estimate the concentration of cfDNA (Appendix A). All cfDNA samples were tested using the KRAS G12/13 screening kit (cat. #1863506, BioRad, Hercules, CA, USA) regardless of the *KRAS* mutation status of the PMs. The performance of all kits was verified using gBlocks when available (Integrated DNA Technologies, Iowa, USA); 191–230nt fragments of synthetic double-stranded DNA containing the nucleotide change of interest (Appendix A).

### 2.5. Statistical Analysis

The association between clinico-pathological variables and ctDNA detection was tested using the Fisher’s exact test for two dichotomous variables or the Mann–Whitney U test for a continuous variable combined with a dichotomous variable. Comparison of preoperative cfDNA input to ddPCR reactions was performed using the Mann–Whitney U test. Comparison of preoperative cfDNA input with follow-up samples was tested using the Kruskal–Wallis test for a continuous variable in multiple groups. Statistical significance was defined as a *p*-value < 0.05 (two-sided test). Univariate associations between DFS and clinico-pathological variables and ctDNA detection were tested using the log-rank test (Kaplan–Meier method). A cox regression analysis was performed to generate hazard ratios and 95% confidence intervals (95% CI). No correlation between OS and clinico-pathological variables was calculated as the number of events within the follow-up period was insufficient for statistical analysis. Dichotomization was performed on the basis of mean values for continuous variables. Statistical analyses were performed using the Statistical Package for Social Sciences (SPSS) version 23 for Windows (IBM Corporation, Armonk, NY, USA) and GraphPad Prism v7.02 (GraphPad Software Inc., San Diego, CA, USA).

## 3. Results

### 3.1. Patient Baseline Characteristics

Thirty patients eligible to receive CRS-HIPEC were included in the study, following the exclusion of fourteen patients for the reasons outlined in Figure 1. Baseline characteristics of all patients are presented in Table 1. Of these patients, 24 underwent CRS-HIPEC after intraoperative assessment determined them to have resectable metastases. None of the patients had received neo-adjuvant chemotherapy prior to CRS-HIPEC. To account for the effect of liver metastases on ctDNA levels, CRS-HIPEC patients were postoperatively sub-classified into those with isolated PMs (*n* = 22) or PMs with resectable liver metastases (*n* = 2) (Figure 1). The remaining six patients were intraoperatively disqualified from CRS-HIPEC so instead underwent an open–close procedure due to a PCI higher than 20 (*n* = 4), or because irresectable liver metastases (*n* = 1) or para-aortic lymph nodes (*n* = 1) were discovered.

### 3.2. Accuracy of ctDNA Analysis for Detection of PMs

Between one and four mutations (median 1.5) were identified in the tumor tissue of PMs from each of the 30 patients in the study. The three most frequently mutated genes were: *KRAS* (18/30 patients; 60%); *TP53* (13/30 patients; 43%) and *APC* (8/30 patients; 27%). Mutations in these genes are typically found in 43%, 60% and 81% of non-hypermutated CRC tumors, respectively [42]. A detailed list of mutations is described in Appendix A. By targeting these tissue-guided mutations by ddPCR analysis, ctDNA was detected preoperatively in 10/30 (33%) patients (Figure 2A). In the subgroup of patients who underwent CRS-HIPEC, the rate of detection was the same (8/24; 33%) and was marginally lower when the two patients with liver metastases were excluded (6/22; 27%). To estimate the relative amounts of ctDNA shed into the plasma, a median VAF of 1.8% (range 0.6–10) was determined for the eight patients who had detectable ctDNA. Interestingly, the VAF was marginally higher in patients who were later diagnosed with a systemic recurrence (median 2.8%, range 2.1–10, *n* = 4) compared to a loco-regional recurrence (median 0.6%, range 0.6–1.4, *n* = 3, *p* = 0.057). In comparison, patients in the open–close subgroup had the same detection rate as the overall study group (2/6; 33%), with a median VAF of 1.75% (range 0.9–2.6, *n* = 2). This suggests that there is no correlation between the preoperative PCI and the likelihood of ctDNA detection in the circulation.

Detection of ctDNA did not correlate significantly with any of the tested clinical variables used to assess eligibility for CRS-HIPEC, including PCI (Appendix A). When the total cfDNA concentration was calculated for all 30 patients, there was no significant difference between samples that contained detectable ctDNA and those that did not (*p* = 0.422), or between samples from patients who underwent CRS-HIPEC (median 8.7ng ml^−1^, range 4.3–70.4) or an open–close procedure (median 7.2 ng ml^−1^, range 4.4–9.9; *p* = 0.174) (Figure 2C). If a cfDNA sample contained multiple mutations, their proportions closely mirrored those observed in PMs (Figure 2B), which suggests a faithful representation of tumor DNA in the circulation. All preoperative plasma samples had 100% concordance with the *KRAS* mutation status in PMs when tested with the KRAS G12/13 screening kit, which further indicates that the PMs were the source of the ctDNA.

### 3.3. Preoperative ctDNA as a Prognostic Marker of Recurrence

The association between preoperative ctDNA detection and recurrence was tested in 14/24 patients who were diagnosed with a recurrence during the follow-up period (median DFS 17 months, range 6–25). The proportion of these patients who had detectable ctDNA preoperatively was higher for those with a systemic- (4/5, 80%) compared to a loco-regional recurrence (3/9, 33%). Regardless of the type of recurrence, preoperative ctDNA detection was associated with a median DFS of 6.0 months (95%-CI 1.8–10.2), significantly worse compared to patients without ctDNA detection (median DFS not reached, *p* = 0.016; HR 3.454, 95% CI 1.145–10.423) (Figure 3A). When the two patients with resectable liver metastases were excluded from the survival analysis, both of whom had detectable ctDNA, a trend was still observed, but the difference was no longer significant (median DFS 7.0 months vs median DFS not reached, *p* = 0.086; HR 2.673, 95% CI 0.806–8.857) (Figure 3B). A univariate analysis was performed to test the association between other clinically relevant clinico-pathological variables and DFS after CRS-HIPEC. A PCI higher than 10 (5.0 months versus median DFS not reached, *p* = 0.035) and the presence of liver metastases (2.0 months versus 12.0 months, *p* < 0.001) were found to have a significant association (Appendix A).

### 3.4. ctDNA to Support Recurrence Diagnosis During Follow-up

None of the samples taken initially after CRS-HIPEC had detectable levels of ctDNA, except for patient L-27. In this case, recurrences to the lungs and the spleen were diagnosed 7 months after CRS-HIPEC, which suggests the potential presence of systemic micro-metastases at the time of CRS-HIPEC. Due to the exploratory approach of this study, only 19 of the 24 patients provided additional samples during the follow-up period. Of the five patients who had a systemic recurrence, ctDNA was detected in four out of four patients who provided a follow-up sample (Figure 4). Of the nine patients who had a loco-regional recurrence, ctDNA was detected in one out of eight patients who provided a follow-up sample. Detection of ctDNA in these samples either occurred at or after diagnosis of a recurrence. Circulating tumor DNA was not detectable in any of the seven patients who provided a sample and did not have a recurrence during the follow-up period, which suggests a negative predictive value. Notably, levels of background cfDNA were significantly higher during the 2 months following CRS-HIPEC (*p* ≤ 0.001) and returned to preoperative levels by 7 months, commensurate with tissue damage associated with the procedure (Appendix A).

## 4. Discussion

This pilot study demonstrates the promising feasibility of ctDNA as a prognostic marker of recurrence in CRC patients with PMs who are eligible for CRS-HIPEC. Preoperative detection of ctDNA could influence the decision to undergo CRS-HIPEC but larger studies are required to validate the clinical utility of this approach. In this study, ctDNA was detected preoperatively in 33% of patients and there was a trend between detection and reduced DFS. However, it was not possible to quantify the extent of PMs based on ctDNA detection and cfDNA quantification. This is the first study to apply preoperative tissue-guided ctDNA analysis exclusively to patients selected for CRS-HIPEC to treat PMs of CRC.

A comparable study by Baumgartner et al. (2018) used an NGS-based approach to investigate preoperative ctDNA in patients who underwent surgery to treat PMs of various cancer types. This study reported an overall ctDNA detection rate of 39% across all the included cancer types, with a solid-tissue concordance of 35.3% when comparison was possible [30]. In our investigation, the ctDNA detection rate in patients with isolated PMs was similar at 33%. Despite patients having a form of metastasis, these ctDNA detection rates are more in line with those seen in stage I CRC (40%) [43]. There is little evidence that ctDNA detection in early-stage CRC has prognostic value [28], but interestingly in our study it was found to be predictive of a reduced DFS, despite a comparatively low detection rate. PMs are understood to spread to the peritoneum through a local form of dissemination [2,44], rather than through the circulation [45]. The metastatic site is an important determinant of ctDNA detection, as CRC metastases to the lung, for example, have a lower VAF than to liver or lymph nodes [46]. If CRC patients have isolated PMs, the VAF tends to be lower compared to patients with no PMs or PMs with involvement from other metastatic sites [45]. We hypothesize that ctDNA from isolated PMs is usually poorly shed into the plasma compared to primary or systemically-spread CRC, which could explain the comparatively low preoperative ctDNA detection rate and lack of a significant relationship to the PCI score. Preoperative detection of ctDNA could occur because these patient’s PMs shed greater than usual quantities of ctDNA and are more prone to systemic spread. This might also explain why seven out of eight patients who had detectable ctDNA preoperatively were later diagnosed with a recurrence, despite the intended removal of all PMs by CRS-HIPEC. Alternatively, there may have been an additional ctDNA contribution from un-diagnosed systemic metastases that already existed below the detection threshold of imaging techniques at preoperative assessment. CtDNA from these metastases may have been masked in the weeks following CRS-HIPEC, due to increased background cfDNA from tissue damage and inflammation [47], then become detectable again at diagnosis of recurrence. Another reason for postoperative detection could have been the presence of minimal residual disease after CRS-HIPEC. Recent studies have shown patients with localized CRC to be at a higher risk of recurrence if ctDNA was detected after resection [48]. Similarly, locally advanced rectal cancer patients could be stratified into groups at high or low risk of recurrence based on postoperative ctDNA detection [49].

If preoperative ctDNA detection is validated as a marker of a more invasive tumor type or of undiagnosed metastases, the use of a local treatment such as CRS-HIPEC as a stand-alone therapy may have limited curative potential in this situation. A more appropriate treatment for these patients would be (neoadjuvant) systemic therapy in addition to CRS-HIPEC, an approach used in the PRODIGE-7 trial [14] and is currently under investigation in the CAIRO-6 study [50]. It may even be appropriate to withhold CRS-HIPEC, as the median DFS of 6 months in patients with detectable preoperative ctDNA is similar to the typical physical and quality of life recovery time [51].

Our results with a limited number of available follow-up blood samples hints at the potential of ctDNA analysis to support recurrence monitoring after CRS-HIPEC. Detection of ctDNA in samples from all four patients with a systemic- and one out of eight with a loco-regional recurrence, either at or after the diagnosis of recurrence by standard techniques, suggests a confirmatory role for ctDNA. If validated by a larger study, this finding could lead to an earlier start of palliative treatments to reduce disease-related symptoms, or indicate a repeat HIPEC with curative intent in patients with oligo- and loco-regional recurrence [18].

Despite the limited number of tissue-guided mutations tested in cfDNA by ddPCR, the detection rate was comparable to Baumgartner et al. (2018) who screened for mutations with an NGS panel. We identified a median of 1.5 mutations per patient (range 1–4, *n* = 30) in PMs, with all tested mutations found back in samples that contained ctDNA, except for in one follow-up sample from patient L-18. Baumgartner et al. (2018) detected a median of 2 mutations (range 1–6, *n* = 7) in ctDNA of CRC patients with PMs [30]. A comparable study of ctDNA in patients with metastatic or recurrent CRC had a similar detection rate (median 2 mutations, range 0–25, *n* = 74) [52]. An explanation for our slightly lower detection rate could be that the TSACP panel was limited to 48 genes, so may have missed less common mutations. This, plus the low sample numbers, could also explain the difference between the mutations detection in our study and those detected in frequently mutated genes described in the literature. A limitation of our tissue-guided approach is that patients who had not undergone a primary tissue resection before CRS-HIPEC—in this case 5/30 patients—would need a biopsy of PMs to enable ctDNA analysis. However, available biopsied tissue does not guarantee that mutations will be identified, as three patients were excluded from the study due to a lack of targetable mutations in their PMs. If biopsied tissue is not available or mutations are not detected, an NGS panel of commonly mutated genes should be considered to screen for ctDNA, provided a sensitivity comparable to ddPCR can be reached.

An alternative approach to non-invasive PM assessment is through improved imaging modalities. Two recent studies showed the feasibility of DW-MRI to improve the detection and quantification of colorectal PMs compared to conventional CT imaging [25,53]. However, DW-MRI could underestimate the extent of metastases, especially of signet ring cell or mucinous adenocarcinomas [25,53]. In our study, 9/30 (30%) patients had these sub-types so may not have benefitted from DW-MRI alone. Interestingly, a study by Vidal et al. 2017 showed patients with mucinous tumors to have a lower than expected ctDNA VAF [45]. This poses an additional challenge for the detection and quantification of tumors with mucinous histology. Molecular characterization of the tumor through a solid or liquid biopsy, as demonstrated in our study, would still be required to enable the use of targeted therapies, for example. Confirmatory studies are necessary before DW-MRI can be implemented as a standard technique for analysis of PMs.

The clinical variability of patients in our pilot study supports the feasibility of our approach. Inclusion of patients with a primary tumor in situ or liver metastases reflects the clinical reality of CRS-HIPEC candidates, even though these factors may have increased the chance of ctDNA detection. To the best of our knowledge, this is the first study to analyze the mutation status of PMs and both pre- and postoperative plasma samples in a cohort of CRC patients eligible to receive CRS-HIPEC. Investigations are needed to understand the biological factors affecting ctDNA representation in plasma, particularly in cases of localized dissemination such as PMs. Although a pilot study, the findings presented here are potentially practice changing and should be validated by larger clinical studies.

## 5. Conclusions

This pilot study demonstrates the feasibility of ctDNA as a prognostic marker in the clinical management of CRC patients with PMs. If ctDNA is detected preoperatively, patients may experience greater benefit from chemotherapy in addition to, or instead of, CRS-HIPEC. Additionally, the approach outlined here could support the detection of recurrences along-side conventional diagnostic methods during follow-up. To allow clinical implementation, these results require confirmation by larger trials and ultimately, by prospective studies in which treatment decisions are based on ctDNA analysis.

## Figures and Tables

**Figure 1 jcm-09-01738-f001:**
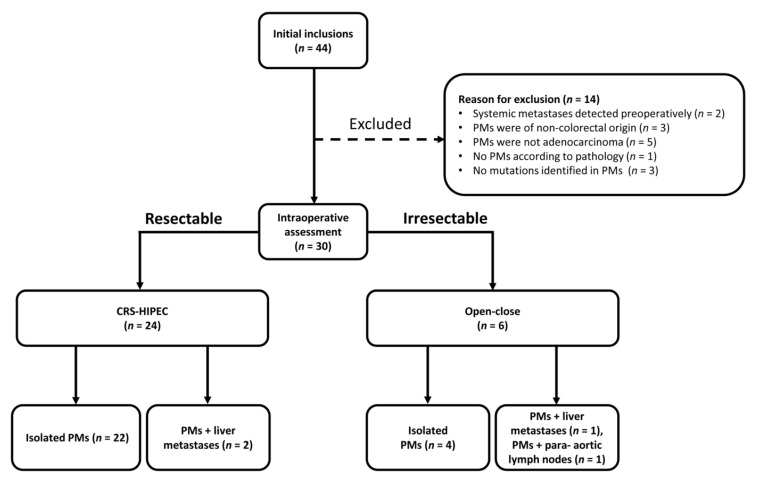
Flowchart of patient classification after initial inclusion in the study. The reasons for patient exclusion prior to intraoperative assessment are described. After the intraoperative assessment, patients either underwent Cytoreductive Surgery and Hyperthermic Intraperitoneal Chemotherapy (CRS-HIPEC) or received an open–close procedure. PMs: Peritoneal Metastases.

**Figure 2 jcm-09-01738-f002:**
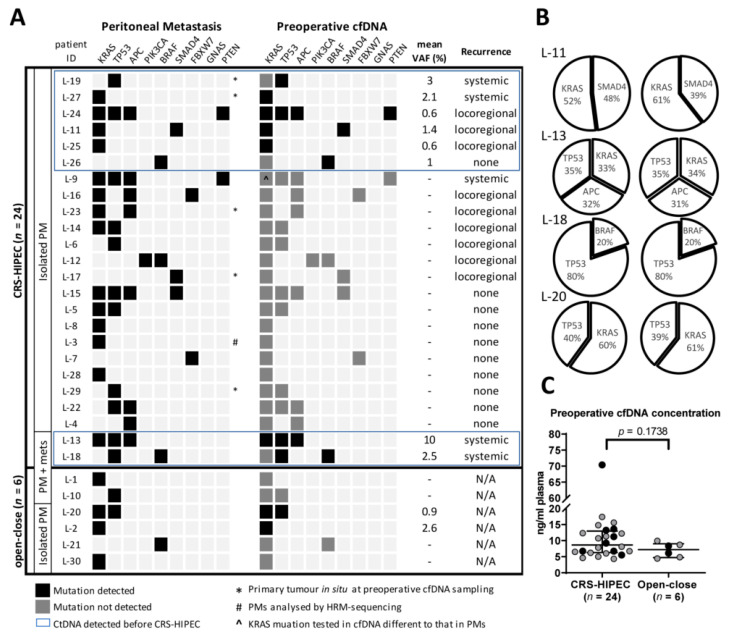
(**A**) An overview of the genes in which a somatic mutation was identified in peritoneal metastases (PMs) and tested for in the corresponding preoperative cfDNA sample. A black square indicates that the mutation was identified in both PMs and cfDNA; a grey square indicates a negative result for the corresponding mutations in the cfDNA. All cfDNA samples were tested using the KRAS G12/13 screening kit. (**B**) Comparison of the relative proportion of mutations detected in PMs and preoperative cfDNA samples. (**C**) Concentration of cfDNA samples. A black dot denotes that ctDNA was detected; lines indicate the median and the interquartile range. HRM-sequencing: High-Resolution Melting assay followed by Sanger sequencing.

**Figure 3 jcm-09-01738-f003:**
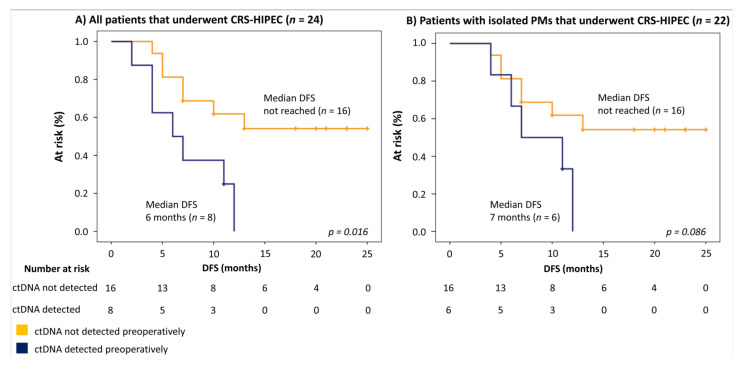
Survival analysis of patients who received CRS-HIPEC. (**A**) All patients (*n* = 24). (**B**) All patients except those diagnosed with preoperative liver metastases (*n* = 22). DFS: Disease-free Survival.

**Figure 4 jcm-09-01738-f004:**
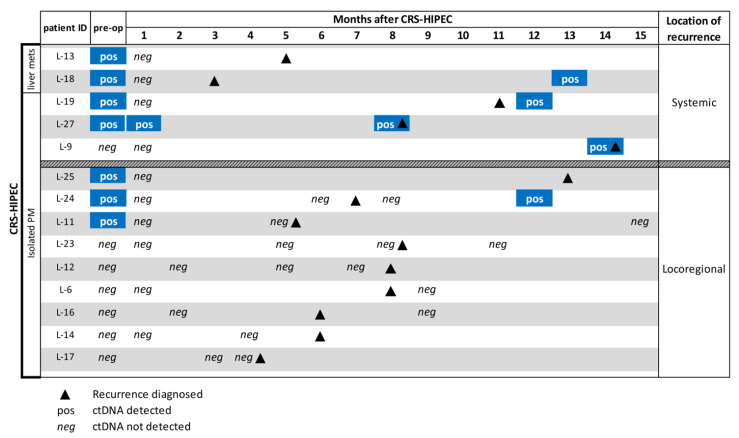
Tissue-guided ctDNA analysis of plasma samples taken from 14/24 patients during follow-up after CRS-HIPEC. Ten patients did not provide a sample. Patients are categorized by the location of metastases before or during CRS-HIPEC and the location of recurrence(s) during follow-up. The blue color is to highlight a positive result.

**Table 1 jcm-09-01738-t001:** Baseline characteristics of all patients in the study and the selection of patients that underwent Cytoreductive Surgery and Hyperthermic Intraperitoneal Chemotherapy (CRS-HIPEC).

		All Patients (*n* = 30)		Underwent CRS-HIPEC (*n* = 24)	
**Characteristics**					
**General**					
Age in years	Mean/SD	65.1	9.4	65.2	9.6
	Median/range	67	37–81	66.5	36–81
Male gender	(*n*/%)	17	56.7	14	58.3
BMI	Mean/SD-all	27.1	5.2	27	5.5
	male	25.6	3.1	25.4	2.7
	female	29.1	6.7	29.2	7.6
	Median/range-all	26.8	21.3–49	26.8	21.3–49
	male	24.4	21.3–32.1	24.3	21.3–29.8
	female	28.4	22.0–49.0	27.9	22.0–49.0
ASA classification	I-II (*n*/%)	22	73.3	17	70.8
	III (*n*/%)	8	26.7	7	29.2
**Primary Tumor**		*n*	%	*n*	%
Location	Colon	27	90	22	91.7
	Rectum	3	10	2	8.3
TNM-stage at diagnosis	II	8	26.7	7	29.2
	III	10	33.3	8	33.3
	IV	12	40	9	37.5
Differentiation grade	Good/moderate	24	92.3	19	90.5
	Poor	1	3.8	1	4.8
	Signet cell	1	3.8	1	4.8
Lymph invasion		7	25.9	5	22.7
Venous invasion		10	37	8	36.4
Tumor type	Adenocarcinoma	21	70	17	70.8
	Mucinous adenocarcinoma	8	26.7	6	25
	Signet cell type	1	3.3	1	4.2
**Treatment**					
Adjuvant chemotherapy primary	(*n*/%)	11	36.7	9	37.5
Primary tumor in situ at intended CRS-HIPEC	(*n*/%)	8	26.7	7	29.2
Liver metastases at intended CRS-HIPEC	(*n*/%)	3	10	2	8.3
Lymph node metastases at intended CRS-HIPEC	(*n*/%)	7	24.1	6	25
PCI	Mean/SD	10.1	8.4	9.3	7.8
	Median/range	7	0–31	7	0–26

SD: Standard Deviation. BMI: Body Mass Index. ASA: American Society of Anesthesiologists Physical Status Classification System. TNM: TNM classification of malignant tumors. PCI: Peritoneal Cancer Index.

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
