# Peer review of "Circulating Tumor DNA as a Preoperative Marker of Recurrence in Patients with Peritoneal Metastases of Colorectal Cancer: A Clinical Feasibility Study"

_jcm, 2020, doi:10.3390/jcm9061738_

Round 1

Reviewer 1 Report

The authors should be commended for carrying out an interesting pilot study with potential clinical utility pending further research. Some recommendations are outlined below:

Introduction

-The data regarding CRS and HIPEC for peritoneal metastases has been controversial in colorectal cancer. I think it needs to be made clearer that this is only a viable option in high volume specialist centres.

-Line 56: I would argue that CEA is not a routine diagnostic test in all CRC patients. It is most often used as a surveillance to detect early relapse. It cannot differentiate between liver mets vs peritoneal metastases etc and so I would re-phrase 'can also be used to detect PMs'

Study design and patients

-I think a CONSORT diagram should be included as it is not clear how many patients were identified between August 2016-March 2018 and how many were then excluded to get to the 30 included patients. Although exclusion criteria are listed, it is not clear how many patients were excluded for each of the reasons described.

Blood and tumour tissue collection

-Further detail on this is provided in the supplemental methods where it says that blood samples were either collected in Streck tubes or EDTA tubes. Is it really fair to introduce variability like this particularly given the small number of included patients? It states that all samples were centrifuged on the day of collection. However, for EDTA samples to be viable for downstream ctDNA analysis, samples need to be centrifuged within 2-4 hours after collection. It is well described in the literature that pre-analytical factors are of paramount importance.

Results

-Baseline characteristics table 1 (please make it clear that TNM stage is at diagnosis)

-Line 180: the KRAS mutation rate appears a little higher than expected and the APC appears lower than expected and so I would suggest that it is inaccurate to state ‘at similar frequencies typically seen in CRC’

-Figure 2A: it might be good to add the post-operative cfDNA data here as well since recurrence is included

-Figure 3:  Is it really appropriate to have a p value when the number of patients included is so small? What is the hazard ratio and the 95% CI? Where are the numbers at risk?

Discussion

-I think the limitations of this being a hypothesis generating pilot study with a small size need to be emphasized more

-It would be worth discussing the results in the context of the wider ctDNA CRC literature whereby pre-operative ctDNA has not been shown to be a prognostic biomarker in other populations but has been shown to be a marker of minimal residual disease in non-metastatic CRC following resection.

-The studies by Vidal et al 2017 and Bando et al. 2019 whereby ctDNA was less likely to be detectable with peritoneal metastases or a mucinous histology could be referenced.    

Author Response

Response to reviewer 1

The authors thank the reviewer for their thoughtful feedback. We have addressed the individual suggestions, which are detailed below in red text.

--------------------------------------------------------------------------------------------------------------------------------------

Comments and Suggestions for Authors

The authors should be commended for carrying out an interesting pilot study with potential clinical utility pending further research. Some recommendations are outlined below:

Introduction

  1. The data regarding CRS and HIPEC for peritoneal metastases has been controversial in colorectal cancer. I think it needs to be made clearer that this is only a viable option in high volume specialist centres.

This is indeed a relevant point. We have added text to the introduction to acknowledge that the addition of HIPEC following CRS has not shown significant survival benefit in clinical trials (lines 44-46). We go on to explain that patients in our study did not receive neo-adjuvant systemic therapy prior to  CRS-HIPEC and that a different chemotherapy was used during HIPEC compared to the PRODIGE trial (lines 46-48). To make it clearer that improvements in overall survival are not just due to CRS-HIPEC alone, we explain that it is offered along with other treatments, including neo-adjuvant systemic chemotherapy, at dedicated centres (lines 48-50).

  1. Line 56: I would argue that CEA is not a routine diagnostic test in all CRC patients. It is most often used as a surveillance to detect early relapse. It cannot differentiate between liver metastases vs peritoneal metastases etc and so I would re-phrase 'can also be used to detect PMs'.

We have removed the text that describes the CEA test as part of the diagnostic procedure for all CRC patients. We have also added text to clarify that the CEA test is not able to give site-specific information about metastases (lines 62-64).

Study design and patients

  1. I think a CONSORT diagram should be included as it is not clear how many patients were identified between August 2016-March 2018 and how many were then excluded to get to the 30 included patients. Although exclusion criteria are listed, it is not clear how many patients were excluded for each of the reasons described.

We have adapted figure 1 to include the reasons for patient exclusions, with numbers, so it is clearer how we eventually arrived at 30 inclusions. We have added text to the results section to explain that patients were excluded and we refer the reader to figure 1 for more detail (lines 173-174).

Blood and tumour tissue collection

  1. Further detail on this is provided in the supplemental methods where it says that blood samples were either collected in Streck tubes or EDTA tubes. Is it really fair to introduce variability like this particularly given the small number of included patients? It states that all samples were centrifuged on the day of collection. However, for EDTA samples to be viable for downstream ctDNA analysis, samples need to be centrifuged within 2-4 hours after collection. It is well described in the literature that pre-analytical factors are of paramount importance.

We agree that the type of tube and the time to centrifugation are important pre-analytical factors in ctDNA analysis. We aimed to use Streck tubes to collect every blood sample, but these are non-standard tubes that were not always available to the clinician at the time of blood collection. If streck tubes were unavailable, EDTA tubes were used. We have added text to the supplementary materials and methods to explain this (lines 10-11).

Although there is still debate about the effect of time before centrifugation on ctDNA levels, studies have shown cfDNA concentrations to be stable up to 24 hours after blood collection in EDTA (1, 2). Another study showed there is no difference between the yield from EDTA and Streck up to 48 hours (3). The consensus is clear, however, that blood should be centrifuged as soon as possible after collection. In our study we stated that samples were centrifuged on the day of collection. We have now changed the methods section in the main text (lines 128-129) and the supplementary materials and methods (line 11) to say that all samples were centrifuged within eight hours of collection.

Results

  1. Baseline characteristics table 1: please make it clear that TNM stage is at diagnosis.

Table one has been adjusted so that ‘TNM-stage’ now reads ‘TNM-stage at diagnosis’.

  1. Line 180: the KRAS mutation rate appears a little higher than expected and the APC appears lower than expected and so I would suggest that it is inaccurate to state ‘at similar frequencies typically seen in CRC’.

We agree that this needs clearer explanation. In the results section, we have added the frequencies of mutations in KRAS, TP53 and APC reported in a major study (lines 196-197). In the discussion, we have added text to explain that the reason our detection rate varied from other studies could be due to the TSACP panel we used, which was limited to 48 genes. We also explain that the low sample number could also have contributed to this variability (lines 317-321).

  1. Figure 2A: it might be good to add the post-operative cfDNA data here as well since recurrence is included.

We decided not to include the postoperative ctDNA data in figure 2A because it would have distracted from the message of the figure, which was to demonstrate the utility of ctDNA as a preoperative marker. The postoperative sampling was done at heterogeneous time points so we

decided it could be more clearly displayed in a separate figure (fig 4).

  1. Figure 3: Is it really appropriate to have a p value when the number of patients included is so small? What is the hazard ratio and the 95% CI? Where are the numbers at risk?

The p-value is informative when directly comparing two groups, even when the sample number is small. We feel it would have been strange to omit this. We agree, however, that extra analysis would be beneficial given the sample size, so have performed a cox regression analysis and provided the hazard ratio and confidence intervals for patient sub-groups (patients with or without liver metastases) in the results section (lines 233 and 236). The materials and methods section has been updated to include the cox regression analysis (lines 164-165). We have also added a numbers-at-risk table to figure 3.

Discussion

  1. I think the limitations of this being a hypothesis generating pilot study with a small size need to be emphasized more.

We have added text in multiple places to highlight that this is a pilot study and the findings need to be validated by larger clinical trials (lines 263-266, 339, 345-347).

  1. It would be worth discussing the results in the context of the wider ctDNA CRC literature whereby pre-operative ctDNA has not been shown to be a prognostic biomarker in other populations but has been shown to be a marker of minimal residual disease in non-metastatic CRC following resection.

We have added text and new references to the discussion to address the utility of postoperative ctDNA detection in identifying MRD in non-metastatic colorectal cancer. We go on to explain that MRD may be one of the reasons that postoperative ctDNA was detected in our study (lines 292-296).

  1. The studies by Vidal et al 2017 and Bando et al. 2019 whereby ctDNA was less likely to be detectable with peritoneal metastases or a mucinous histology could be referenced.

We have incorporated in to the discussion the finding by Vidal et al that the location of PMs can affect the ctDNA VAF (lines 280-281). We have also added the finding from the same study that mucinous tumors have a lower VAF than non-mucinous (lines 333-335).

We have included the finding by Bando et al that the metastatic site of CRC can have an important effect on the VAF (lines 278-280). We have also altered the text that stated Baumgartner et al as being the only comparable study, as these references have proven that to be inaccurate (lines 271).

References

  1. van Ginkel JH, van den Broek DA, van Kuik J, Linders D, de Weger R, Willems SM, et al. Preanalytical blood sample workup for cell-free DNA analysis using Droplet Digital PCR for future molecular cancer diagnostics. Cancer Med. 2017;6(10):2297-307, DOI:10.1002/cam4.1184.
  2. Risberg B, Tsui DWY, Biggs H, Ruiz-Valdepenas Martin de Almagro A, Dawson S-J, Hodgkin C, et al. Effects of Collection and Processing Procedures on Plasma Circulating Cell-Free DNA from Cancer Patients. The Journal of molecular diagnostics : JMD. 2018;20(6):883-92, DOI:10.1016/j.jmoldx.2018.07.005.
  3. Kang Q, Henry NL, Paoletti C, Jiang H, Vats P, Chinnaiyan AM, et al. Comparative analysis of circulating tumor DNA stability In K3EDTA, Streck, and CellSave blood collection tubes. Clin Biochem. 2016;49(18):1354-60, DOI:https://doi.org/10.1016/j.clinbiochem.2016.03.012.

Reviewer 2 Report

In this manuscript, the authors propose that Ct-DNA can serve as a marker for systemic metastases in patients with colorectal cancer. The manuscript is well-written and discusses the advantages and limitations of the described Ct-DNA analysis. Publication into the Journal of Clinical Medicine is recommended by this reviewer.

Minor comments:

Is the mismatch repair status of the patients known? Would MMR status affect the results presented in this manuscript?

Authors can provide more detail in Table 1. Due to differences in body fat distribution among the sexes, average and median BMI for females and males should be provided, rather than providing an average BMI value for one sex.   

In Figure 3, the format of the y-axis is incorrect. Values are separated by a comma instead of a period. Format of Figure 3B should be consistent with that of Figure 3A.

Figure 4 contains a green outline around the box that may be an artifact of copy/pasting.

Author Response

Response to reviewer 2

We thank the reviewer for these comments. We have addressed the individual points below in red text.

--------------------------------------------------------------------------------------------------------------------------------------

Comments and Suggestions for Authors

In this manuscript, the authors propose that Ct-DNA can serve as a marker for systemic metastases in patients with colorectal cancer. The manuscript is well-written and discusses the advantages and limitations of the described Ct-DNA analysis. Publication into the Journal of Clinical Medicine is recommended by this reviewer.

Minor comments:

  1. Is the mismatch repair status of the patients known? Would MMR status affect the results presented in this manuscript?

The mismatch repair (MMR) status of the patients is unknown. Testing for MMR status is not part of the standard testing workflow and the patients in this study did not meet the specific criteria to receive the test. If a patient’s PMs were to be MMR-deficient and subsequently hyper-mutated, it is unlikely to have had an effect on ctDNA detection because we use a personalised approach to identify mutations in the patients PMs. There is also no evidence in the literature that MMR-deficient tumors shed ctDNA at a higher rate than non MMR-deficient tumors. We have added text to the materials and methods to say that MMR status was not tested for (line 109-110).

  1. Authors can provide more detail in Table 1. Due to differences in body fat distribution among the sexes, average and median BMI for females and males should be provided, rather than providing an average BMI value for one sex.

We have adjusted table 1 to provide the mean and median BMI categorised by patient sex.

  1. In Figure 3, the format of the y-axis is incorrect. Values are separated by a comma instead of a period. Format of Figure 3B should be consistent with that of Figure 3A.

Commas have been replaced with periods in the y-axis of figure 3.

  1. Figure 4 contains a green outline around the box that may be an artifact of copy/pasting.

We cannot see the green outline that the viewer describes. This is indeed likely to be an artefact of copy/paste or of the PDF generation software. We will send a separate version of figure 4 if requested by the editor.

Reviewer 3 Report

Overall the authors perform the circulating tumor (ct) DNA analysis to select patients for Cytoreductive Surgery and Hyperthermic Intraperitoneal Chemotherapy CRS-HIPE for colorectal cancer patients with peritoneal metastases (PMs).  Author have incorporated   a NGS-based mutation analysis and droplet digital (dd)PCR method to detect mutation from PMs and (cf)DNA. cfDNA samples from 33% of patients are associated with a reduced disease-free survival (DFS) after CRS-HIPEC.

Very nice manuscript, these results indicated that ctDNA can serve as a marker for recurrence in patients with PMs of colorectal cancer.

Nevertheless, the authors should elaborate all the methods they have used in the method section.

Also, check the spelling in the text, there are many mistakes.  

Author Response

Response to reviewer 3

We thank the reviewer for these comments. We are pleased to see that our main findings have been so clearly understood. Please see our response to the individual comments below in red text.

---------------------------------------------------------------------------------------------------------

Comments and Suggestions for Authors

Overall the authors perform the circulating tumor (ct) DNA analysis to select patients for Cytoreductive Surgery and Hyperthermic Intraperitoneal Chemotherapy CRS-HIPE for colorectal cancer patients with peritoneal metastases (PMs).  Author have incorporated a NGS-based mutation analysis and droplet digital (dd)PCR method to detect mutation from PMs and (cf)DNA. cfDNA samples from 33% of patients are associated with a reduced disease-free survival (DFS) after CRS-HIPEC.

Very nice manuscript, these results indicated that ctDNA can serve as a marker for recurrence in patients with PMs of colorectal cancer.

  1. Nevertheless, the authors should elaborate all the methods they have used in the method section.

We have added extra information to the materials and methods section where appropriate (lines 124-125, 135-136, 142-143). We direct the reviewer to the supplementary materials and methods for extra detail.

  1. Also, check the spelling in the text, there are many mistakes.

We thank the reviewer for their careful attention to detail. We have corrected several spelling mistakes throughout the text.

Round 2

Reviewer 1 Report

The manuscript is significantly improved and most of the points have been addressed adequately. There are still a couple of suggestions to highlight as listed below:

-Line 62-63: I still disagree with a statement that suggests that the CEA can be used to detect peritoneal metastases. I assume that authors mean 'the CEA tumour marker can be elevated in the presence of metastases however its sensitivity in the presence of isolated peritoneal metastases is limited (60%)'.

-Please double check that the hazard ratios are indeed correct. See below extract from manuscript line 230. The p value is significant but the CI contains the number 1. The HR is less than one but conventionally it is usually more than 1 when concluding that something is worse than its comparator.

'Regardless of the type of recurrence, preoperative ctDNA detection was associated with a median DFS of 6.0 months (95%-CI 1.8-10.2), significantly worse compared to patients without ctDNA detection (median DFS not reached, p=0.016; HR 0.359, 95% CI 0.108-1.191).'

-In the discussion I can see that the authors have attempted to address my initial recommendation of referring to the wider literature and including the fact that ctDNA has also been shown to be a marker of minimal residual disease in non-metastatic CRC following resection. However, I had also suggested that it would be useful to discuss how in the non-metastatic setting unlike in this study, pre-operative ctDNA does not appear to be a prognostic marker. This has not been addressed.
